# A Formative Study of the Implementation of Whole Genome Sequencing in Northern Ireland

**DOI:** 10.3390/genes13071104

**Published:** 2022-06-21

**Authors:** Katie Kerr, Caoimhe McKenna, Shirley Heggarty, Caitlin Bailie, Julie McMullan, Ashleen Crowe, Jill Kilner, Michael Donnelly, Saralynne Boyle, Gillian Rea, Cheryl Flanagan, Shane McKee, Amy Jayne McKnight

**Affiliations:** 1Centre for Public Health, Queen’s University Belfast, Belfast BT12 6BA, Northern Ireland, UK; katie.kerr@qub.ac.uk (K.K.); caoimhez.mckenna@belfasttrust.hscni.net (C.M.); cbailie06@qub.ac.uk (C.B.); julie.mcmullan@qub.ac.uk (J.M.); acrowe737@qub.ac.uk (A.C.); j.kilner@qub.ac.uk (J.K.); michael.donnelly@qub.ac.uk (M.D.); sboyle52@qub.ac.uk (S.B.); shane.mckee@belfasttrust.hscni.net (S.M.); 2Northern Ireland Genomic Medicine Centre, Belfast Health and Social Care Trust, Belfast BT9 7AB, Northern Ireland, UK; gillian.rea@belfasttrust.hscni.net (G.R.); cheryl.flanagan@belfasttrust.hscni.net (C.F.); 3Regional Molecular Diagnostics Service (Germline), Belfast Health and Social Care Trust, Belfast BT9 7AB, Northern Ireland, UK; shirley.heggarty@belfasttrust.hscni.net

**Keywords:** genomics, rare disease, collaboration, public health, multiomics

## Abstract

Background: The UK 100,000 Genomes Project was a transformational research project which facilitated whole genome sequencing (WGS) diagnostics for rare diseases. We evaluated experiences of introducing WGS in Northern Ireland, providing recommendations for future projects. Methods: This formative evaluation included (1) an appraisal of the logistics of implementing and delivering WGS, (2) a survey of participant self-reported views and experiences, (3) semi-structured interviews with healthcare staff as key informants who were involved in the delivery of WGS and (4) a workshop discussion about interprofessional collaboration with respect to molecular diagnostics. Results: We engaged with >400 participants, with detailed reflections obtained from 74 participants including patients, caregivers, key National Health Service (NHS) informants, and researchers (patient survey *n* = 42; semi-structured interviews *n* = 19; attendees of the discussion workshop *n* = 13). Overarching themes included the need to improve rare disease awareness, education, and support services, as well as interprofessional collaboration being central to an effective, mainstreamed molecular diagnostic service. Conclusions: Recommendations for streamlining precision medicine for patients with rare diseases include administrative improvements (e.g., streamlining of the consent process), educational improvements (e.g., rare disease training provided from undergraduate to postgraduate education alongside genomics training for non-genetic specialists) and analytical improvements (e.g., multidisciplinary collaboration and improved computational infrastructure).

## 1. Introduction

Rare diseases are often debilitating, life-limiting disorders, with varying global definitions averaging a prevalence of 1 in 2500 individuals [1]. They have significant public health implications, cumulatively impacting ~450 million people worldwide [2]. Identifying the number of individual rare diseases is challenging due to issues identifying, recording, and coding rare disease diagnoses; in April 2022 there were 6177 unique ORPHAcodes (Orphanet nomenclature used to code the diagnosis of a rare disease) aligned to ICD-10 codes [3]. The number of rare diseases is increasing, with rare molecular subtypes of common complex disorders also being identified [4]. Rare diseases are not explicitly coded in the UK Health Research Classification System, making it challenging to effectively evaluate research activity within health research portfolios.

Recent advances in multiomic laboratory techniques and complex computational analyses are enabling the identification of more accurate, timely molecular diagnoses. Knowing the molecular cause of a rare disease(s) is essential to facilitate patients equitably benefitting from rapid advances in personalised therapies, such as gene silencing, molecular patches, and gene-editing [5]. Provision of an accurate diagnosis is likely to enable access to relevant support groups, facilitate family planning and appropriate educational/employment support options, and reduce the significant psychological distress experienced by individuals and their caregivers in the uncertainty of a diagnostic odyssey [6,7,8,9,10]. The UK National Health Service (NHS) expenditure on the care and treatment of patients with rare diseases was more than £3.4 billion during the 10 years prior to diagnosis; with an average annual cost per individual of £13,000 [11].

Given that many rare diseases have a genetic aetiology, coupled with the explosion of next generation sequencing technologies seen over the past decade, rare disease diagnostic approaches are now moving towards genome-wide analysis internationally [12,13,14,15,16,17,18]. However, there are significant logistical challenges associated with implementing and delivering an effective, efficient clinical service aligned to rapid research advances in diagnosis and therapy. More than 110,000 people across NI are affected by a rare disease at some point in their lives; [19] the average time to receive an accurate rare disease diagnosis is five years, with half of rare disease patients receiving at least one misdiagnosis [20]. NI has a single regional genetics centre (RMDS), based within Belfast City Hospital. This manuscript presents the results of a study on the introduction of whole genome sequencing (WGS) for patients with eligible rare diseases across Northern Ireland (NI) including an appraisal of local processes to implement WGS.

## 2. Materials and Methods

An overview of the project of implementing WGS in NI is illustrated in Figure 1.

The study comprised an appraisal of the implementation phase of WGS and included eliciting the views of key stakeholders about the project: patients, carers, nurses, doctors, genetic counsellors, clinical scientists, administration staff and researchers. Patients and carers who met the rare disease eligibility criteria for the 100,000 Genomes Project (100 KGP) at the time of recruitment and provided informed consent [21,22] were invited to participate in this research project by their secondary care physician. Samples were shipped to Genomics England Limited (GEL) for WGS, with results returned to NI; multiomic analysis is ongoing.

A key agreed goal was to aim to deliver a comprehensive person-centred service. In keeping with this goal, and mindful of the need to involve stakeholders [23], the research team undertook a formative, developmental study that was iterative and participatory in terms of working with patients, families and advocacy groups and listening to, and learning from their experiences to facilitate the co-evolution of a pragmatic genomic medicine service for patients [24]. Nine community engagement events that incorporated open discussions about the implementation of WGS were held Feburary 2017–March 2018. More than 300 participants attended the events across NI. Furthermore, our research framework for assessing patient-centred care was discussed at a public open meeting and then emerging findings were presented at a second public meeting in 2020. The second public event included presentations from ‘experts by experience’ and round-table discussions (formal notes were not taken from the round-table discussions as this event was not included in our ethical approval). The nature of our research framework and these activities were designed to help build public trust.

### 2.1. Quantitative Analysis of the Implementation and Delivery of WGS as a New Genetic Test Service

We explored processes associated with the identification and recruitment of patients, including informed consent, sample collection and processing, variant interpretation, validation of WGS results, and delivery of primary results to patients and their families where appropriate. Within the cohort who received a genetic diagnosis, the characteristics of participants were extracted and analysed, such as the participant group (i.e., recruited as a singleton or with relatives in a duo, trio, quad or quin), the recruiting medical speciality clinic, and the depth of phenotyping (defined as the number of human phenotype ontology terms reported per participant). Quantitative analysis was conducted using IBM SPSS Statistics 27, complemented by qualitative research with a range of stakeholders. 

### 2.2. Participant Self-Reported Views and Experiences

A survey was iteratively designed with input from two participants, two family carers of participants, researchers with social science, genetic epidemiology expertise, and a clinical geneticist. The first draft was distributed to 10 participants during their recruitment interview and returned at that time (January–May 2018, 22 questions, both open-ended and closed-ended), however, administrative challenges meant that the survey was delayed until June 2020 when it was revised to include several additional questions and posted to a further 400 participants by Belfast Health and Social Care Trust when results were beginning to be returned to participants (32 questions, a stamped addressed envelope was included and a link to the survey online with completion required by 30 September 2020). Respondent data were analysed both quantitatively and qualitatively, and supported by illustrative quotes.

### 2.3. Semi-Structured Key Informant Interviews 

Semi-structured interviews (*n* = 19) were conducted between November 2020 and March 2021 with the NIGMC’s principal investigator (*n* = 1), the project manager (*n* = 1), admin staff (*n* = 1), nurse specialists (*n* = 1), genetic counsellors (*n* = 2), recruiting hospital doctors across a range of medical specialities (*n* = 12), and clinical leads from specialities who expressed an interest in participating in the project but who did not enrol patients (*n* = 1). Interview data were transcribed by interviewers, followed by data familiarisation, coding, theme identification and interpretation [25].

### 2.4. Workshop about Interprofessional Collaboration for Molecular Diagnostics

The service specification also recognised the key need to develop and promote a culture of interprofessional collaboration in order to deliver an improved service for patients and families with rare diseases. Therefore, the research team held a workshop in February 2022 to facilitate discussion and gather data regarding the perspectives of stakeholders about interprofessional collaboration with respect to the specific topic of molecular diagnostics (*n* = 13 participants). The workshop was advertised (via the Northern Ireland Rare Disease Partnership website www.nirdp.org.uk (accessed on 13 June 2022), Twitter, with a post reach of 1773 impressions, and on Eventbrite) and held virtually via Microsoft Teams. Individuals living and working with rare diseases discussed three broad areas: Benefits of interprofessional collaboration on molecular diagnostics and research.Barriers to interprofessional collaboration on molecular diagnostics and research.What should change about collaboration on molecular diagnostics and research.

Contemporaneous notes were taken and data underwent thematic synthesis [25]. 

## 3. Results

### 3.1. Exploring the Logistics of Delivering WGS as a New Genetic Test for NI

A virtual NI genomic medicine centre (NIGMC) was established using funding from the NI Executive and UK Medical Research Council, a funding model that differed from the way in which genomic medicine centres across England have been funded. The NIGMC was launched in 2015, with recruitment beginning in 2017 and results from the initial analysis were reported to patients in 2019. To raise awareness of WGS and encourage more use of genetic approaches within mainstream medicine, training sessions were held with representation from ten medical specialities before recruitment commenced. We undertook an appraisal of local processes to implement WGS for 1073 participants (*n* = 442 probands, no sample fails) who received WGS across eight medical speciality clinics, including processes relating to recruitment and delivery of test results. 

Of the 442 probands, 296 (67%) were recruited by the regional genetics clinic (Table 1), these patients had a higher depth of phenotyping (*p* = 1.96 × 10^−22^) and significantly higher diagnostic yield (*p* = 0.009, OR = 2.1, 95% CI [1.201, 3.797]). No significant difference was observed in the diagnostic yield of participants recruited as singletons compared to duos, trios, quads, or quins (*p* = 0.241). Figure 2 illustrates the main processes and suggestions for improvement as WGS is implemented for future projects, such as the development of a transformational informatics infrastructure-Genomics Open Core Engine for Accelerating Northern Ireland Care (GenOCEANIC). Multiomic analysis has resulted in additional diagnoses with prioritised variants of unknown significance. 

### 3.2. Participant Self-Reported Views and Experiences

The survey had a low response rate overall with 10 participants returning their form in person (100% for the 1st version of the survey), 23 via post, and nine submitting responses online (8% for the revised survey). Respondent demographics were only requested for the revised survey, with this information captured for 32 respondents (Appendix A) [26].

Initial participation: Most respondents (*n* = 37, 92.5%) learned of the project from their doctor; two respondents did not answer that question. One respondent had enquired about any genetics study that they could become part of due to a history of hereditary disease while two respondents heard of the project through “*talks given at a rare disease event*” and subsequently requested to participate. Most respondents confirmed they were happy with their recruitment appointment, with 81% (*n* = 34) explaining they had sufficient time to voice concerns and 73.8% stating they were well explained (*n* = 31). Less than half of respondents (40.5%) accessed the 100,000 genomes website. More information was requested on DNA sequencing, accessing health records, return of results and what was meant by ‘future contact’. Participant concerns included: lack of sufficient information provided; remaining anonymous and the safe storage of their data and results; if the collection of genomic data could impact future life insurance premiums, the result itself, and the potential implications of that on their lives; the implications of a result for their children and grandchildren; the likelihood and accuracy of the result and the length of time it might take to obtain an answer. Participant expectations were primarily for a diagnosis and/or more information on their genetic profile, further understanding of their condition, and peer support, while others expected their data would be used for a better understanding of genetic conditions for future generations:


*“… gain a broader knowledge and was able to see that this project would have such positive outcomes for the future and the identification and treatment of rare diseases.”*
[p22]

Diagnosis and results: From 37 respondents stating their time seeking a diagnosis at the time of recruitment, 25 (67.5%) had been waiting more than five years with 12 (32.4%) waiting more than 10 years. Forty-eight percent of respondents received their results by phone or post as agreed at their recruitment appointment, however, in hindsight 20% of these individuals would have chosen to receive their results differently. Restrictions due to the COVID-19 pandemic also caused challenges in reporting results with many planned in-person appointments cancelled.


*“Obviously COVID-19 knocked things all about. Our child is very precious so to receive first a letter with all the medical diagnosis and language was very unnerving. A phone call or appointment would have been more appropriate, but I know it is strange times… It was very nerve wrecking until I got speaking to [the doctors].”*
[p26]


*“It was an excellent opportunity…Getting the results during COVID19 via letter was very disheartening.”*
[p26]

All respondents would have liked to receive a formal written document of results, information about their disease or follow-up investigations, associated implications for themselves and their family, links to online support, latest research and upcoming events (Figure 3): 


*“Would be great to have information instead of google…”*
[p20]

Eighty-six percent of respondents would prefer direct access to their genetic data as part of a patient portal to help manage complex healthcare needs. Thirty-nine respondents rated their understanding of -omics terms used in the discovery of some disease genes. While many respondents reported a basic or comfortable knowledge of the term genetics (89.7% *n* = 35), and to a lesser degree the terms genomics (53.8%, *n* = 21), WGS, multiomics, transcriptomics and epigenomics were not familiar to most, (Figure 4).

A rare disease focused information hub for NI was proposed in 2013, [7,27] and is now an explicit commitment in the 2022 NI rare disease action plan; [28] respondents indicated the hub should primarily include details on specific rare diseases; latest developments and treatment options; clinical trial information; links to support groups for patients, families, and carers; links to clinical experts, opportunities for peer support; self-help downloadable leaflets and signposting to services, such as benefits advice and transition services.

When asked, ‘What could we do better’ in regards to this project, the majority of respondents were very positive, for example feeling *“very blessed”* and *“pleased”* with *“helpful and caring staff”* and good access to information *“I had lots of technical questions that the genetic counsellor (recruiter) could not answer about the 100 K genomes project. But they directed me where to get more information: [named person] who was excellent”*.

Some respondents were initially happy and excited to be part of the project but were disappointed with the long wait and no contact between the time of recruitment and receiving results. Several participants (35%) shared a more unsatisfactory experience of participating in the project giving reasons such as having no follow-up communication [by the time of the survey], being unable to obtain any updates or information when they looked for it, and not obtaining any answers about their condition from their results, (Figure 3). 

Suggestions for improvement include more regular follow-ups, more information during recruitment and the delivery of results, direct access to results, opportunity for results in writing and talking to a doctor, more realistic timeframes, some success stories shared, a seminar/webinar of the project with updates, a newsletter/email or website updates; improved communication overall was requested by the majority of respondents (53%).


*“I believe the more information and knowledge I have to help my child, helps more than not knowing at all and only working on scraps of information”*
[p28]

### 3.3. Semi-Structured Interviews of Healthcare Professionals Who Participated in WGS

Nineteen semi-structured interviews of key informants for WGS were conducted, including HCPs, lead investigators for the project, genetic counsellors, specialist nursing staff and administrative leads. Four central themes were identified (associated codes are in Appendix A):Healthcare professionals had a positive experience with WGS.Facilitating WGS was a significant workload burden.Interviewees found that participants expressed some concerns about additional findings and time to results.There is a need for additional training.

Theme 1—HCPs had a positive experience with WGS: HCPs understood that WGS will soon become a much larger part of patient care, and therefore, were keen to learn and understand more about this type of care and the benefits it can bring (Figure 5). Most interviewees (89.5%) reported they had a positive experience and expressed their willingness to be involved if a similar project was to take place in the future. They emphasised the benefit of WGS in ending the diagnostic odyssey, helping patients feel there is work being conducted to support them, and modifying a patient’s defined care pathway:

Additionally, HCPs reported that they found the multi-disciplinary approach to be useful for interpreting WGS results:


*“I think that the multi-disciplinary team meetings with the lab were brilliant… I do think that kind of lab liaison is helpful… both in terms of understanding the variant interpretation and being able to explain that to patients. It was helpful for the whole clinical process and from an educational point of view, and even just people working together better.”*
[p14]

Theme 2—Participating in WGS was a significant workload burden: Interviewees frequently commented that managing participation was difficult on top of their routine workload (Figure 5). This was a consistent theme reported by clinicians, specialist nurses and administrative support staff alike, with the need to protect time for future projects, such as these and expand the workforce emphasised: 


*“We can’t multiply people overnight… One of the great outcomes [for the NIGMC] is the upskilling and the education and training of their staff… so there’s a lot of good, but … there’s a lot of unknowns that should be flagged up in the future for any major new service.”*
[p9]

Recruiters discussed the need for a more streamlined consent process, which would facilitate participant understanding and minimise confusion, with suggestions for patients to receive literature associated with the project in advance. The importance of spending time with patients to explain the project, the consenting process and making information as understandable as possible cannot be underestimated. Several interviewees emphasised some conversations take longer than others as patients may not speak English as their first language, may not have been considered for genetic testing previously, or be comfortable with the internet to obtain further information. Most of the HCPs interviewed do not routinely pass information to patients, however, this varied depending on the condition. Interviewees expressed difficulties accessing appropriate information for particularly rare genetic conditions and explained that patients are often very proactive in such situations and would frequently find information on social media pages. P18 said that from their experience, patients learn in different ways; using multiple media resources, such as providing visuals, text messages, and links to videos enables people to access information easily on their phones. They are helpful as the more information is given to patients, the better they may appear to be engaged at the appointment and perhaps they will think of many questions when they go home; printed literature or information would be helpful. 

Theme 3—Interviews found that participants expressed some concerns about additional findings and time to results: Generally, interviewees commented that they felt participants had a good understanding of both the project and the consent process. The concerns they did have related to the potential impact of additional findings, time to results and the likelihood of results, though this usually did not prevent them from participating (Figure 5). HCPs also expressed that the participants were concerned with the amount of time it would take for them to receive a result from WGS and stressed that they were constantly trying to manage expectations about this. Some HCPs did express concern that the consent process did not fully prepare participants: 


*…” I do wonder if the consent form and our discussion with patients will truly have prepared them for the incidental findings… to be honest, I don’t know how well they did understand what they were signing up for. But as a doctor, I think we saw that it was overwhelmingly positive and worth it for them. And the only really realistic, next step in terms of testing.”*
[p4]


*“There were some patients who were like, “I don’t really care what I’m consenting for but I just desperately want an answer for my child.” I was having to say, “but I need to go through this with you.” A very typical comment would have been, “Oh, anything I can do to help your research doctor.” And yes, there could be fruitful answers out of it, hopefully. Sometimes they were just so desperate to be eligible. I think for me the biggest thing was managing people’s expectations of a diagnosis at the end of the project.”*
[p10]

Theme 4—There is a need for additional training: Multidisciplinary collaboration was an essential component of this project, but the need for additional training was stressed, with several interviewees highlighting the importance of integrating WGS into undergraduate medical training as the future of medicine (Figure 5). Several respondents explained they struggled to understand the concept of genetics and were concerned that, because of their lack of knowledge, they do a disservice to patients by not making more frequent referrals to genetics. By being involved in this project they hoped they would become more familiar with genetics and gain confidence in what tests and processes are available. Several educational sessions were held with all staff involved in this project invited with individual follow-up for specific medical specialities; all participants confirmed receiving excellent support from helpful project staff, but many interviewees said they struggled to remember details of training when there was a lag between training and recruitment.


*“…having something you can actually go back to refresh your memory could be really helpful”.*
[p18]


*“Sometimes you get, you get everything’s taught at the same time and then it’s gonna be ages before you actually get to use that information. And then you kind of forget.”*
[p14]


Other important findings from the interviews


Several additional key findings were identified following the interviews, which did not fall under the above specific themes. 

The potential of exploring lower tiered variants, as interpretation generally focused on the top 1 and 2 tiered variants from the Genomics England bio-informatics pipeline, was highlighted as an area where additional diagnoses were potentially being missed:


*“I think it’s a tier three that’s probably more niggling to myself and my colleagues, because we realise a large portion of our patients are waiting from results specifically out of tier 3 and we haven’t got the resources or man power yet to address those, and I think that is where the problem potentially lies.”*
[p16]


*“We have found that whatever way the algorithm’s working, that a fairly high percentage of the time our actual results, that causative result, is in Tier 3. OK, definitely not the majority of the time, or else in the un-tiered, but it could be as high as 5–10% of the time.”*
[p4]

In addition, the challenges of travelling to clinics for recruitment to the 100 KGP were discussed as a particular barrier for patients with rare diseases in NI, especially for those managing a disability or balancing child-care: 


*“[Patients] had to come up to Belfast for appointments, which was probably a problem for the patients, especially if they had any disabilities or children, then just getting them up for travel. And the appointment times started at 9:00 AM in the morning, if they had other children to get out to school or anything.”*
[p12]

Interviewees were very supportive of an informational website for individuals with rare diseases in NI to refer to, suggesting it should contain the following information: A simple approach to rare disease information that both non-specialists and health care professionals could access easily.Financial entitlements that patients are entitled to with relevant links.Online consenting for research, perhaps via a video call.Signposting facility to avoid the website being too heavy on information, e.g., to patient support groups.Disease specific information.Educational resources regarding genetic/genomic testing.Contact details of willing clinicians working in particular areas of rare disease.

### 3.4. A Discussion Workshop on Interprofessional Collaboration for Molecular Diagnostics

Thirteen participants attended a 1-h workshop, from a variety of backgrounds including individuals with rare diseases, their relatives, clinicians, academic researchers, industry representatives and technical staff. Following data familiarisation and coding, five themes were identified (associated codes are available to view in Appendix A):Resource constraints prevent collaborative rare disease research.Collaborative rare disease research is hindered by ineffective communication.Rare disease awareness, support and information services are insufficient.Current administrative systems are barriers to collaborative rare disease research.Interprofessional collaboration is beneficial for rare diseases.

Theme 1—Resource constraints prevent collaborative rare disease research: The benefits of technology for facilitating remote rare disease research were discussed, however, for research that must be conducted in person, a lack of accessible venues for people with a range of disabilities was stressed:


*“Venues need to be accessible to all. …[often] accessible facilities aren’t truly accessible.”*
[p1]

The lack of funding and trained researchers available to conduct rare disease research in NI was also discussed, with an emphasis on how research is not perceived as encouraged in multiple areas of clinical practice:


*“Research is not encouraged in NI clinical practice and a lot of people in areas of rare disease have lost their opportunity to be involved in research. It’s difficult to motivate training clinicians to be involved in research.”*
[p2]


*“We need to not only encourage people to do research but also show them a pathway to make this research applicable to their career, e.g., is there funding for a post that then continues [their research]? How does this augment my career?”*
[p3]


*“It’s very difficult to bring in money to do this type of research, money is so competitive.”*
[p1]


*“We need more rare disease researchers and more investment in rare disease research in NI”*
[p4]

In addition, the importance of rare disease education throughout all levels was considered:


*“[Rare disease awareness] starts from primary school, if you have somebody who has a rare disease, if you have teachers that are made aware of it, it can grow and mushroom in the community.”*
[p1]

Theme 2—Collaborative rare disease research is hindered by ineffective communication: Ineffective communication was discussed from several angles including a lack of advertisement of research participation opportunities and dissemination of results. 


*“There is a lot of these workshops and surveys but there isn’t a great amount of progress from it, what happens from this research?”*
[p1]


*“My only experience of a rare disease research project was as a patient, signing a consent form prior to a surgery and then never hearing anything else from it. I want to know what comes out of the research projects!”*
[p3]


*“You don’t hear any of the results that’s coming out of *[research projects]*, especially true in rural areas…We need more information to be posted on social media. You don’t hear about project progress either.”*
[p4]


*“You can’t take part if you don’t know about research”*
[p1]

Clear methods of communication with HCPs about the management of an individual’s rare disease were also highlighted as lacking: 


*“Seeing the consultant is great and getting the letter is great, but if you have no idea how to contact them back, people fall apart. Having some way of being able to contact them back and *[knowing]* what do we need to do next is crucial.”*
[p3]

Theme 3—Rare disease awareness, support and information services are insufficient: A lack of rare disease awareness and emotional support, both generally in the community and amongst HCPs was stressed. In addition, the need for a central rare disease information hub was discussed: 


*“We don’t have a *[rare disease]* centre, it would be nice to have this so that families or people that are affected could meet and discuss mental health issues.”*
[p1]


*“Emotional support for this type of research is really crucial, need for reassurance, there may be anxiety about taking part in studies, so there needs to be somewhere they can go ask questions, could possibly work with charities to do this.”*
[p6]


*“It’s not the healthcare professionals’ fault that they don’t always know about all rare diseases, we need to increase awareness amongst them too.”*
[p1]

Theme 4—Current administrative systems are barriers to collaborative rare disease research: The need for a streamlined ethics application protocol across multiple hospitals in NI was emphasised strongly by academic researchers and HCPs alike, in particular with regards to clinical research:


*“Still waiting on approval for minor [ethics] revisions (sitting on it for 3 months). With [University] institutions you can easily monitor the progress of ethics applications, but with the hospitals they go into a black hole. It’s very hard to get feedback.”*
[p3]


*“… the biggest challenge is the ethical approval, it takes significantly longer than any other region…”*
[p2]

Theme 5—Interprofessional collaboration is beneficial for rare diseases: Despite the challenges and barriers, participants were strongly in favour of collaborative research: 


*“In England they had a collaborative approach, we were able to produce research mostly by systematic reviews which was able to shape the R21 pathway. It was really nice to be part of a team that translated *[research]* into progress.”*
[p2]


*“A multidisciplinary approach was beneficial, (including scientists, midwifes, clinical geneticists) to discuss the case and interpret the evidence.”*
[p2]


*“Maybe there’s one person in NI who has a rare disease, that person may only be able to go to America to find out something about their rare disease. Therefore, benefits of collaborative research do outweigh the challenges.”*
[p1]


*“Rather than trying to reinvent the wheel, especially North and South of the *[Irish]* border, we only have a certain number of specialists in this field (particularly in genomic specialist researchers), so there needs to be better collaboration.”*
[p2]

## 4. Discussion

This evaluation of NI introducing WGS for rare diseases provides reflections from a range of stakeholders to inform future research and clinical practice, with emphasis on administrative, analytical, and communication elements (Figure 6). The difference in the funding model between NI and, for example, England which had far greater financial resources, and therefore, more capacity for recruitment.

### 4.1. Administrative Considerations for WGS Rare Disease Diagnostics in a Regional Setting

While patients with rare diseases and their relatives were generally positive about receiving WGS (even in cases where no genetic explanation was found for a patient’s condition), problems with communication were consistently highlighted. A named point-of-contact with whom patients and their caregivers communicate for regular status updates, as well as general timeframes for various elements (primary analysis, secondary analysis, cascade testing, etc.), would be helpful, as would a project website with general information. Many participants waited approximately three years for the return of results, a significant delay due to the logistical difficulties in delivering WGS and worsened by the COVID-19 pandemic. Analytical time is becoming shorter as improvements to genomic healthcare are implemented, with recent findings from the Newborn Sequencing in Genomic Medicine and Public Health study in the United States of America providing a diagnosis for infants from WGS in less than five days [29], and rapid genomic sequencing being offered to critically ill neonates in the UK through, for example, the Great Ormond Street Hospital for Children NHS Foundation Trust (GOSH), with other UK clinics now offering similar services [30].

Both HCPs and WGS recipients reported minimal concerns about participation, except for a small number of individuals who expressed concerns about additional/secondary findings and data usage. Healthcare professionals reported that upon further discussion, patients received clarity on additional findings and were generally happy to proceed with WGS, being able to make informed decisions as to whether they would like to know about additional findings. Recent research by Hart et al. (2020) found that despite having initial patient concerns regarding secondary findings from WGS data, participants generally did not report increased anxiety and distress because of them, and individual attitudes towards secondary findings are fluid [31]. Therefore, participants who initially refused disclosure of additional findings should be made aware of the mechanisms to change their decision. There remains significant room for improvement in explaining WGS in future projects, a fact which has also been identified in previous studies of patient/family WGS understanding [32]. With regards to data security, participants were particularly worried that their genomic data may be shared with insurance companies and could ultimately impact their individual premiums as new symptoms, diagnoses, screening or treatments must be disclosed when applying for new insurance policies [33]. Yet these concerns were generally trumped by the individual’s desire to receive a diagnosis for them or their child and were not a preventative barrier to participation. Powell et al.’s (2022) report emphasises the importance of including parental perspectives on the use of WGS for diagnosis, to evaluate what information should be returned and ultimately increase public trust in molecular diagnostics [34]. 

The method of returning a diagnosis varied, with several participants reporting that they would prefer to change how they received their results if they had a second opportunity. Therefore, the mode of results reporting should be carefully considered in the future return of genomic results. Telehealth initiatives can alleviate some of the time burdens; high patient satisfaction levels have been reported following live-video consultations for genetic results [35]. Given that many patients with rare diseases have severe mobility restrictions, attending appointments in person can be very difficult. Live-video conferencing could reduce the number of late cancellations or no-shows which could ultimately decrease patient wait lists. It is also important to note that given results were returned during the COVID-19 pandemic, during which many services were severely curtailed. Patients with rare diseases are at increased risk of premature mortality from COVID-19 [36] and effective telehealth initiatives were the only viable option to return results to many individuals. 

Key informants of the 100 KGP stressed they often encountered difficulties obtaining informed consent from participants, emphasising a need for streamlined administrative support. HCPs expressed a desire for sufficient, protected time allocated for appointments to give HCPs the opportunity to explain WGS to patients/family members and ensure they have a good understanding. This was supported by reflections from the participants, with approximately 1/5th of participants reporting that they did not feel the project was clearly explained to them. In future projects, HCPs could be guided by the 12 key considerations about WGS developed by the Clinical Sequencing Exploratory Research (CSER1) Consortium, which includes the type of tests available, limitations of WGS, clinical implications, emotional implications and implications of results for family members [37]. Although many of the specialties interviewed had not reported results to patients at the time of interview, suggestions were made to provide further guidance around this process to increase HCPs confidence and knowledge when relaying this information to patients. One key administrative challenge was the multiple resources for patient information. GenOCEANIC is a transformational informatics infrastructure which may help address some of these issues by providing a mechanism for clinicians to securely submit patient phenotypic data to inform genetic testing, clinical decision support to assist clinicians, clinical geneticists and laboratory scientists to streamline the processes involved in genomic testing, provide support for molecular multidisciplinary team meetings to help turn sequencing variants into validated results for patients, and link into both Encompass as NI’s forthcoming digital integrated care record [38], and the developing Northern Ireland RAre Diseases & Congenital Abnormalities Registry (NIRADCAR).

### 4.2. Analytical Considerations for WGS

A diagnosis was observed for ~18.6% of probands on first pass analysis, with no difference in diagnostic yield identified in participants recruited as singletons compared to those with any number of relatives. The cost of running a WGS pipeline (from DNA extraction, to WGS and ultimately variant interpretation) was estimated to be £7050 for rare diseases, with 15% focused on the variant interpretation process [39]. It is vital to identify methods of streamlining the WGS variant interpretation process.

Deep phenotyping is crucial to provide a diagnosis from genetic data [40,41,42]. Given that phenotyping guided the selection of the analysis panels, comprehensive phenotyping is critical to identifying a patient’s molecular diagnosis and aid future re-analysis for participants as our understanding of genotype-phenotype associations improve. Participants recruited by the genetics clinic were almost twice as likely to receive a diagnosis compared to all other medical specialities, which may have been impacted by the significantly higher average number of human phenotype ontology terms reported by genetics clinicians prior to WGS. The importance of recording detailed phenotypic information must be emphasised in non-genetics specialist training.

A recurring theme from HCP interviews was the significant burden that participating in this WGS project added to routine workload. As the integration of genomic medicine into routine healthcare grows, this is one area where regulated interprofessional collaboration between clinicians, academic researchers and industry may be of significant benefit. Indeed, several clinicians acknowledged that an individual’s diagnosis may lie beyond the top tiered variants, but that they simply did not have the time to explore these. One example of the benefit of collaboratively exploring lower tiered variants was reported by the respiratory Genomics England Clinical Interpretation Partnership, who found a tier 3 variant causative of hereditary haemorrhagic telangiectasia in a family who participated in the 100 KGP, through further functional multiomic analysis [43]. Further multiomic functional analyses of rare diseases, conducted collaboratively between clinicians and researchers, following recommendations for application of PS3/BS3 ACMG criteria, is vital to fully explore phenotypically plausible variants which may not always be feasible to explore in a clinical environment [44]. Several recent studies have highlighted the power of multiomics for improving our understanding of the biological mechanisms of disease where genetics alone is insufficient, for example in common complex diseases such as hypertension and diabetes mellitus [45,46] as well as rare diseases [47,48,49]. A multidisciplinary, collaborative approach exploring additional variants based on prior genotype-phenotype relationships helps maximise diagnostic yield from WGS [50]. Advances in clinical decision support tools, such as machine learning/artificial intelligence models, will also facilitate variant classification and accelerate rare disease diagnoses [51,52]. 

Several direct-to-consumer genomic testing providers exist which offer whole genome sequencing, for example Dante Labs (https://dantelabs.co.uk/) (accessed on 13 June 2022), Veritas (https://www.veritasint.com/) (accessed on 13 June 2022) and Nebula Genomics (https://nebula.org/) (accessed on 13 June 2022) for ~£500–£1500. It is important to note that these costs do not always include sample validation, detailed VUS interpretation with functional interrogation, re-analysis, exploration of non-coding elements, or clinical sequencing validation of prioritized variants. Furthermore, the UK Science and Technology Committee report on direct-to-consumer genomic testing (2021–2022) highlights that several significant issues still need to be addressed to minimise the risk to consumer. For example, most providers self-certify the safety and efficacy of their services, and as such there is a need for external assessment prior to release on the UK market. The project discussed in this manuscript is a transformational NHS research project aimed towards implementing genomic medicine into mainstream clinics, which critically depends on a standardised recruitment pathway where the patient is clearly identified, comprehensively phenotyped at the time of sample collection, clinical grade sample handling to ensure no sample handling errors or contamination, standardised wet-lab and bioinformatic pipelines applied to sequencing data and multidisciplinary team meetings to discuss variant pathogenicity, validation and agreed mode of results reporting.

### 4.3. Educational Considerations for WGS

Whilst it was encouraging to see that most participants felt they had at least a basic understanding of genetics, it was interesting to note that 15.6% of respondents reported that they had never heard of the term genomics and 6.3% reported they had never heard of whole genome sequencing (WGS), given that genomics and WGS should have been discussed during the initial recruitment appointment. Work is needed to develop resources that clearly explain the fundamentals of genomic medicine to patients and healthcare professional, ensuring they can provide informed consent [53]. This is particularly important for WGS as the potential for identification of secondary findings is not an issue faced by routine healthcare investigations, and participants must clearly understand the difference between the possibility of additional findings from WGS compared to a traditional targeted genetic test. Many participants would like to learn more about multiomic analysis. One effective method of improving public genomics education may be the incorporation of effective genomic curricula at secondary education level, such as the bioinformatics based laboratories trailed by Martins et al. (2020), which crucially also included support for teachers [54]. For healthcare professionals, there is a clear need for training of non-genetics specialists on genomic medicine to provide an integrated mainstream genomic medicine service, with regular refresher training given the field’s dynamic nature. Successful examples of genetics education for non-genetics specialists have been seen through a mixture of online and face-to-face learning on the topics of recognising potentially genetic conditions, discussion of genomics with patients, understanding the next steps in the referral process and improved co-ordination with health care professionals [55]. In addition, the inclusion of laboratory and research rotations has previously been shown to be effective in improving genomics education in genetic counsellors, which should be expanded to include education on multiomic analyses in the future [56]. To further assist with this, it would be useful to develop a streamlined pathway to easily inform HCPs what tests and processes are available, as well as where they can obtain more training if required. 

Patient reported experience and outcome measures (PREMs and PROMs) for people with rare diseases are vital to ensure we are effectively capturing their experiences. These remain largely disease specific and do not represent collective rare diseases, despite the many challenges this group has in common [8,57]. A scoping review by Whittal et al. (2021) discusses many of the challenges faced in capturing PREMs and PROMs for heterogenous rare diseases and provides potential strategies to overcome these, such as collaboration with patient advocacy groups as well as clinical care networks and international data collection to increase the size of data collection [58]. 

### 4.4. Strengths and Limitations

A key strength of this study was the representation of perspectives from a wide variety of stakeholders, with valuable commitment observed in the semi-structured interviews and discussion workshop in particular. Public engagement with this project has been considerable with good attendance in open meetings. Learning from this study is focused on a single nation, and informed the development of a NI Genomics Partnership model [28], to help deliver an effective integrated molecular diagnostics service. Limitations include a low response rate for the postal participant survey, which may be due to the postal survey being circulated for more than a year, in some cases up to 3 years, following sample collection and initial recruitment. It should be noted that all participants who were physically handed the first survey had a 100% participation rate, suggesting that this is the most effective way to collect survey data in future projects, perhaps at regular time points (recruitment/at the return of results). For many patients who will not have heard any further updates from the project, or received a negative result, it would be understandable if participants were disillusioned with the project, and therefore, unwilling to participate. Given the lower than ideal response rate, the patient data may not be fully reflective of the views of participants, but recurring themes were observed and yielded valuable insights. 

## 5. Conclusions

This study presents vital insights from a variety of stakeholders on WGS from experiences in NI, including recommendations for future projects considering administrative, analytical and educational components. Whilst participants were overwhelmingly in favour of WGS for improving rare disease outcomes, there is a clear need for improved awareness and education surrounding rare diseases and molecular diagnostics. Further research using multiomics and in-depth computational analysis is proving essential to refine putative disease-causing variants, thereby increasing the diagnostic yield and facilitating clinical decision making [48,52,59,60,61,62]. Interprofessional, multidisciplinary collaboration between healthcare professionals, clinical/biomedical laboratory scientists, academia, industry, bioinformaticists/data scientists, and patient representatives is essential for optimised rare disease research and clinical practice. 

## Figures and Tables

**Figure 1 genes-13-01104-f001:**
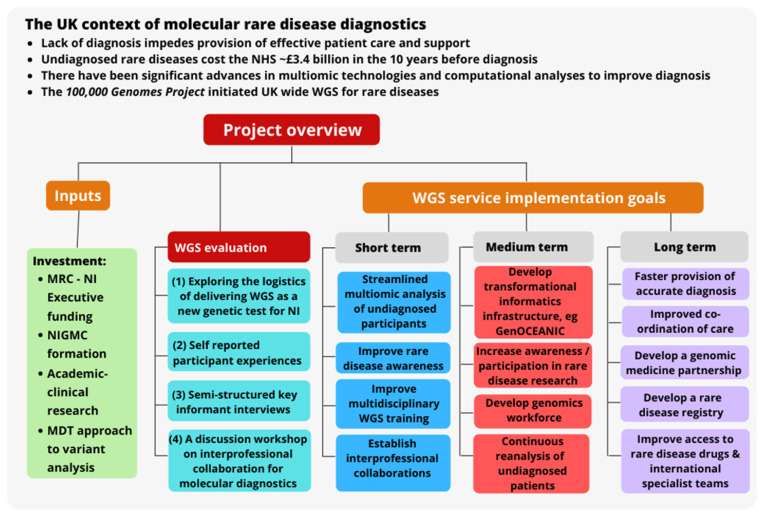
Project implementation overview. Abbreviations: multidisciplinary team (MDT), medical research council (MRC), National Health Service (NHS), Northern Ireland (NI) whole genome sequencing (WGS), United Kingdom (UK).

**Figure 2 genes-13-01104-f002:**
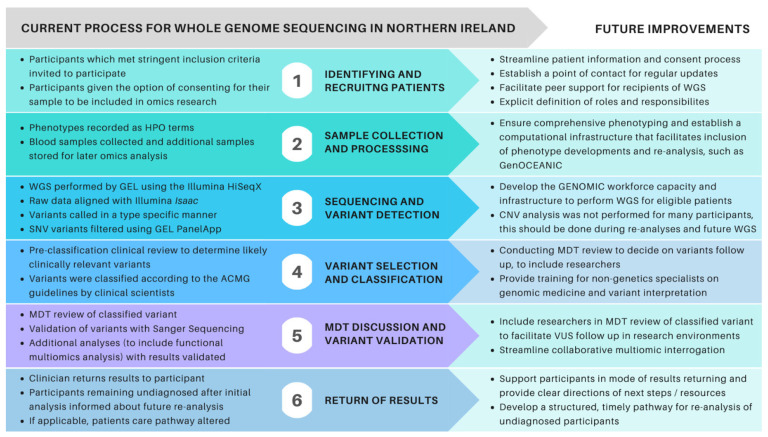
Summary of the workflow and recommendations for future implementation of WGS in NI. Abbreviations: American College of Medical Genetics (ACMG), Copy number variants, (CNV) Genomics England (GEL), multidisciplinary team (MDT), variant(s) of unknown significance (VUS), whole genome sequencing (WGS).

**Figure 3 genes-13-01104-f003:**
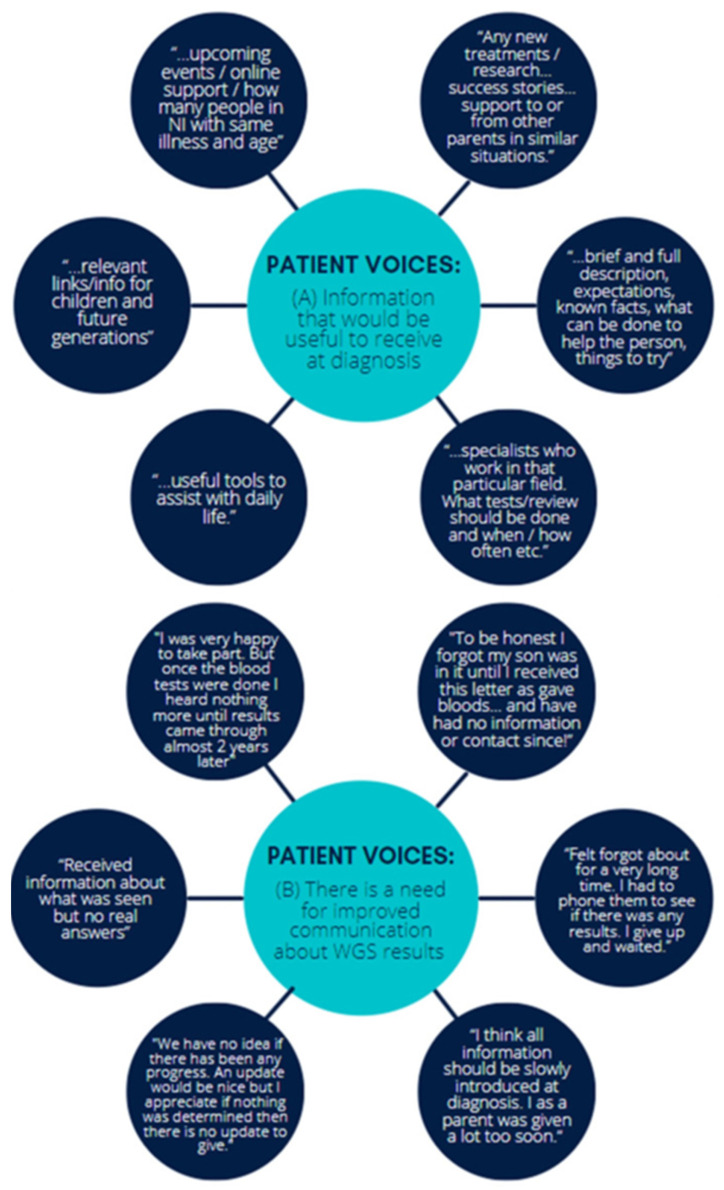
Patient voices about (**A**) what information would be useful to receive at diagnosis and (**B**) the need for improved communication regarding WGS results.

**Figure 4 genes-13-01104-f004:**
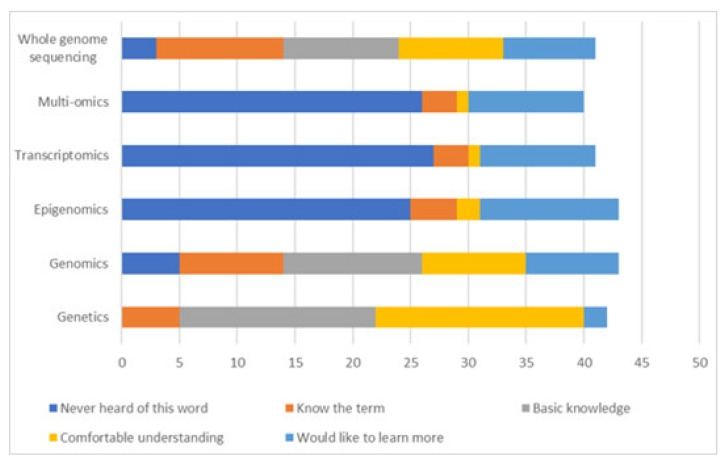
Participant understanding of multiomics terminology.

**Figure 5 genes-13-01104-f005:**
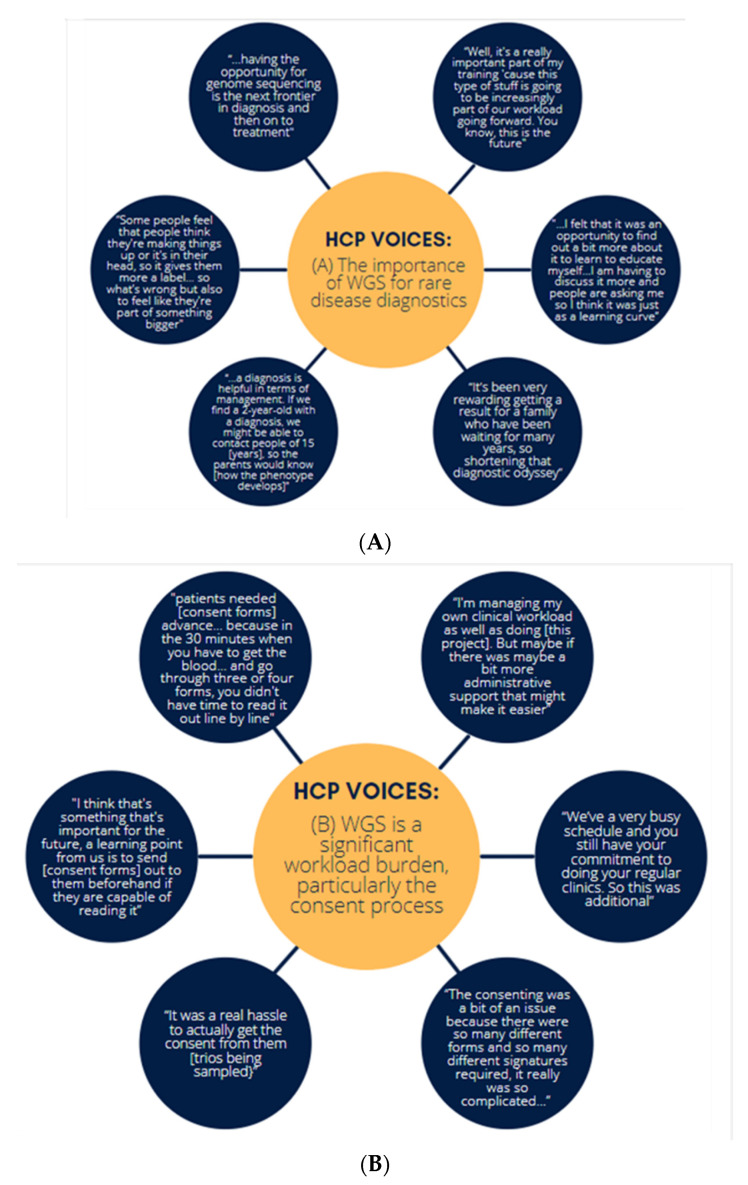
HCP voices about (**A**) the importance of WGS for rare disease diagnostics, (**B**) workload burden of WGS and difficulties in the consenting process, (**C**) participant concerns regarding WGS and (**D**) the need for additional training on molecular diagnostics for rare disease.

**Figure 6 genes-13-01104-f006:**
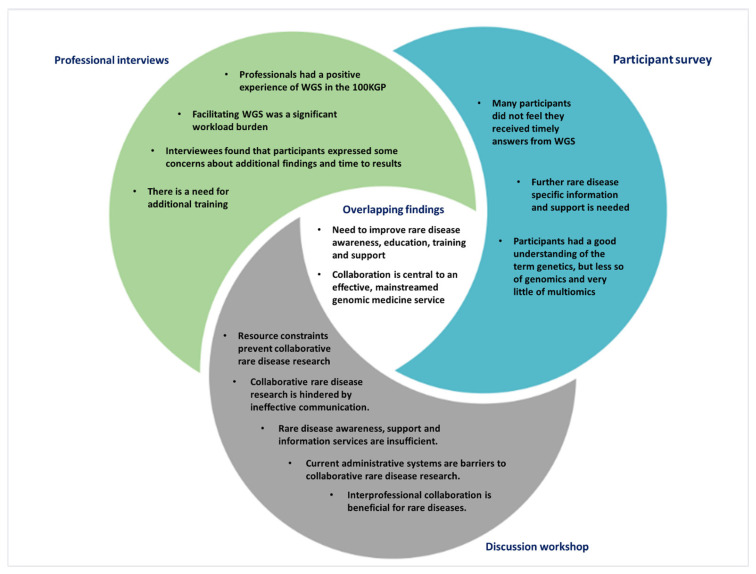
Summary of central themes identified from survey, interviews, and workshop.

**Table 1 genes-13-01104-t001:** Characteristics of probands.

Participant Characteristic	Count (Total n = 442)	Percentage
- Sex (Male/Female)	245/197	55%/45%
Age at recruitment (2018) - Mean (range)	21.4 years (0.5–91)	Not applicable
Recruiting clinic - Genetics - Nephrology - Neurology - Paediatric pathology - Other *	296 41 23 71 15	67.0% 9.3% 5.2% 15.2% 3.4%
Recruitment type - Singleton (proband) - Duo (+1 relative) - Trio (+2 relatives) - Quad/Quin (3 or 4+ relatives)	120 51 250 21	27.2% 11.5% 56.6% 4.8%

* Five or less participants individually, including rheumatology, metabolic, endocrine, and respiratory clinics.

## Data Availability

Not applicable.

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
