# Peer review of "A Formative Study of the Implementation of Whole Genome Sequencing in Northern Ireland"

_genes, 2022, doi:10.3390/genes13071104_

Round 1
Reviewer 1 Report
Understanding benefits, perceptions, and barriers to implementation of whole genome sequencing for rare diseases diagnosis from both a patient and health care provider’s perspective is crucial. The current study provides important insight into the experience of implementing WGS in Northern Ireland. Overall, the manuscript is well written, the methods and results are described well, and the conclusions and recommendations of the authors are supported by the data. A few minor edits are recommended to improve the overall manuscript.
Figure 3 is a major component of the manuscript, however, when viewed in printed color format it is difficult to read due to the small size of the text. This may be amplified if the paper is printed in grayscale or black and white. I suggest increasing text size or editing length of the quotes for easier readability.
Throughout the manuscript the authors should ensure that acronyms are spelled fully out at least the first time they are used. For example, MDT, HCP, MRC, etc.
Please edit for minor grammatical errors and consistency throughout. For example, the use of periods at the end of sentences in lists, etc.
Reviewer 2 Report
The paper by Kerr et al describes an implementation of WGS screening in NI from several years ago. While it's clearly written, I'm not sure it fits the scope of the journal, as there is no gene- or genome-level data. The experience shared here would definitely benefit administrators of analogous enterprises but would likely be of little value to the research-orientated readership. A few points on the contents: 1. The costs seem excessive. Dante Labs provide WGS for some $600, a huge saving. Especially given the ultimate outcome - a letter, which seems to scare some patients, rather than educate them. 2. Logistics seems to be a hurdle. The example of Dante, whereby a saliva sample is self-collected and shipped by a customer, may be educational here. These aspects need to be discussed.
Author Response
See the attachment (reviewer 2)

Round 2
Reviewer 2 Report
no other comments